# ZeroFL: Efficient On-Device Training for Federated Learning with Local Sparsity

**Xinchi Qiu[1,\*], Javier Fernandez-Marques[2,\*], Pedro P. B. Gusmao[1], Yan Gao[1], Titouan Parcollet[3] and Nicholas D. Lane[1]**

[1] Department of Computer Science and Technology, University of Cambridge
[2] Department of Computer Science, University of Oxford
[3] Laboratoire Informatique d'Avignon, Avignon Université

## Abstract

When the available hardware cannot meet the memory and compute requirements to efficiently train high performing machine learning models, a compromise in either the training quality or the model complexity is needed. In Federated Learning (FL), nodes are orders of magnitude more constrained than traditional server-grade hardware and are often battery powered, severely limiting the sophistication of models that can be trained under this paradigm. While most research has focused on designing better aggregation strategies to improve convergence rates and in alleviating the communication costs of FL, fewer efforts have been devoted to accelerating on-device training. Such stage, which repeats hundreds of times (i.e. every round) and can involve thousands of devices, accounts for the majority of the time required to train federated models and, the totality of the energy consumption at the client side. In this work, we present the first study on the unique aspects that arise when introducing sparsity at training time in FL workloads. We then propose *ZeroFL*, a framework that relies on highly sparse operations to accelerate on-device training. Models trained with ZeroFL and 95% sparsity achieve up to 2.3% higher accuracy compared to competitive baselines obtained from adapting a state-of-the-art sparse training framework to the FL setting.

## 1 Introduction

Despite it being a relatively new subfield of machine learning (ML), Federated Learning (FL) (McMahan et al., 2017; Reddi et al., 2021; Horvath et al., 2021) has become an indispensable tool to enable privacy-preserving collaboratively learning, as well as to deliver personalised models tailored to the end-user's local data and context (Arivazhagan et al., 2019; Hilmkil et al., 2021; Cheng et al., 2021). For example: next-word prediction (Hard et al., 2018), physical activity detection (Doherty et al., 2017), keyword spotting (Hard et al., 2020), among others.

Unlike standard centralised training, which normally takes place on the Cloud and makes use of powerful hardware (Hazelwood et al., 2018), FL is envisioned to run on commodity devices such as smartphones or IoT devices often running of batteries, which are orders of magnitude more restricted in terms of compute, memory and power consumption (Qiu et al., 2021). This triplet of factors drastically limits the complexity of the ML models that can be trained on-device in a federated manner, ceiling their usefulness for the aforementioned applications as a result. In order to adjust the memory and compute footprints of complex ML model to the FL setting, the research community has presented a number of approaches including: the use of distillation (Hinton et al., 2015) to enable the aggregation on the server side of heterogeneous model architectures (e.g. based on the compute capabilities of each device) that collaboratively train a single global model (Lin et al., 2020; Zhu et al., 2021); group knowledge transfer algorithm (He et al., 2020); federated dropout, by which clients perform local training on a sub-model of the global model (Caldas et al., 2019), translates into lower overall communication costs and, enables better support for heterogeneous pools of clients regardless of their compute capabilities (Horvath et al., 2021); and, more generally, better aggregation strategies that enable faster convergence (Li et al., 2018; Reddi et al., 2021), reducing in

---

\*Equal contribution. Correspondence to Xinchi Qiu (`xq227@cam.ac.uk`) or Javier Fernandez-Marques (`javier.fernandezmarques@linacre.ox.ac.uk`).

this way overall device utilization (e.g. fewer local epochs) and number of communication rounds. Other optimization techniques such as quantization and sparsity have been used in the context of FL but mostly as a way to reduce communication costs (Liu et al., 2021; Amiri et al., 2020; Shahid et al., 2021) but not to accelerate on-device training.

The use of sparse operations (e.g. convolutions) at training time has recently been shown to be an effective technique to accelerate training in centralised settings (Sun et al., 2017; Goli & Aamodt, 2020; Raihan & Aamodt, 2020). The resulting models are as good or close to their densely-trained counterparts despite reducing by up to 90% their FLOPs budget and, resulting in an overall up to $3.3\times$ training speedup. Acceleration is achieved by performing sparse convolutions during the forward and/or backward pass, which requires at least one of the operands (i.e. inputs, weights, gradients) to be sufficiently sparse and, software and hardware support for such operations. However, it is unclear how the different FL-specific challenges (i.e. data imbalance, stateless clients, periodic aggregation) will restrict the quality of the global model.

This work considers the challenges and opportunities of inducing high levels of sparsity to accelerate training on-device for FL workloads, and provides the following contributions:

- The first framework for Federated Learning that leverages sparsity as a mechanism to accelerate on-device training by inducing up to 95% sparse weights and activations. This work considers three popular datasets: CIFAR-10 and FEMNIST for image classification and, SpeechCommands for audio classification.

- A study on the unique aspects that arise when introducing sparsity at training time in FL: the degree of overlap between non-zero values decreases with layer-depth index and, the locations of zero-valued weights in the global model remain constant throughout most of the training rounds. Our discussion sets the foundations for future research in this area.

- A technique that alleviates the accuracy degradation when applying a state-of-the-art off-the-shelf sparsification method to the FL domain. *ZeroFL* achieves +2.3% and +1.5% higher accuracy than baselines when inducing 90% and 95% sparsity respectively. In addition, ZeroFL also leverages sparsity when transferring the local models to the central server reducing communication costs by $3.0\times$ while still outperforming competitive baselines.

## 2 RELATED WORK

Pruning neural networks involves discarding parts of the model (e.g. individual weights or entire channels) that are *irrelevant* for solving the task at hand. This procedure generally produces a lightweight model representation more suitable for deployment on constrained devices with limited memory and compute budgets. In this section we detail how different forms of pruning or sparsification have been used to accelerate inference and, to a lesser extent, training. We also discuss how these have been introduced to reduce communication costs in distributed and federated learning.

**Unstructured pruning.** Frameworks relying on unstructured pruning (Han et al., 2015a;b; Guo et al., 2016; Molchanov et al., 2017) often achieve higher compression ratios at the expense of inference stages being as compute intensive in practice as those of the original model. This is because, assuming pruning has been homogeneously applied on the model, sparse operations can only be efficiently accelerated on supported hardware, such as modern GPUs (Wang, 2020; Zachariadis et al., 2020; Hong et al., 2018) or custom accelerators (Zhang et al., 2016; Lu et al., 2019; Srivastava et al., 2020), for a sufficiently high sparsity ratio. The lower the ratio, the less likely sparse operations would translate into measurable speedups. In the case of CPUs, speedups due to sparse operations where one operand is unstructurally sparse are often only feasible at 90% sparsity ratios or higher (Hong et al., 2019; Wang, 2021).

**Structured pruning.** Methods that apply structured pruning (He et al., 2018; 2017; Jian-Hao Luo & Lin, 2017; Yu et al., 2018; Molchanov et al., 2019; Wang et al., 2017), on the other hand, trade compression for acceleration potential. These approaches modify the underlying computational graph by discarding entire channels, resulting in smaller but still dense convolution operations, or by removing the nodes all together if an entire layer is set to be removed by the chosen pruning strategy. As a result, structured pruning frameworks are the preferred option when aiming to accelerate inference on general purpose hardware. A body of work across structured and unstructured pruning

methods, attempts to induce structure in otherwise randomly sparse networks S. Gray & Kingma (2017); Ren et al. (2018); Wen et al. (2020); Verelst & Tuytelaars (2020). This is often referred to as *block sparsity* and consists in subdividing the matrix representations of inputs or weights into tiles (e.g. $16 \times 16$ tiles), and restrict the training in such a way that some tiles contain only zeros while the rest remain dense and real-valued. Matrix-matrix multiplications following such a pattern can be accelerated at lower global sparsity ratios compared to those following unstructured sparsity Hoefler et al. (2021). Other forms of constraining how sparsity occurs have been proposed, for example, a cache-aware reordering on the sparsity pattern of the weights Elsen et al. (2020). This can be used to ensure high cache reuse on Cortex-A mobile CPUs, resulting in $2.4 \times$ acceleration of MobileNets.

**Sparse training.** The majority of works making use of sparsity are envisioned for either model compression or to accelerate inference. Only recently, sparse operations have been considered to accelerate training. The work of Sun et al. (2017) presented a mechanism to induce high levels of sparsity in the gradients during backpropagation and, demonstrated large speedups when training MLP-only models. More recently, Goli & Aamodt (2020) build upon the observation that gradients from consecutive batches are near identical. They present a framework to reuse a random sample of previously computed gradients and their thresholded difference w.r.t gradients from the current batch, resulting in a sparse tensor. Their framework accelerates training of CNNs by performing sparse convolutions during the backward pass at the cost of pre-computing partial gradients during forward pass. Closer to our work is SWAT (Raihan & Aamodt, 2020), a framework that relies on sparsified weights during inference and sparsified weights and activations for backward propagation.

**Compression on communication.** Konečnỳ et al. (2016) proposed to restricts the updates of weight matrices to have a pre-specified structure in order to reduce the total communication cost. The structure can either be random or low-rank structure. ATOMO (Wang et al., 2018) introduced a generalised gradient decomposition and sparsification technique, aiming to reduce the gradient sizes communicated upstream. Han et al. (2020) proposed a different way of aggregation in the server, which instead of aggregating model weights, it aggregates the sparsified gradients after every local update step. However, since the method requires to aggregate sparsified gradient after every step, it cannot benefit from multiple local updates. Hence it might require extra communication rounds to reach the target performance. PruneFL Jiang et al. (2019) reduced both computation and communication overhead to minimize the overall training time by including an initial pruning at one selected client and further pruning as a part of FL process.

Nevertheless, none of the aforementioned works explored the challenges of extending state-of-the-art sparsification methods to federated learning as a way to accelerate on-device training. With ZeroFL, a framework specifically tailored to the FL setting, achieves better accuracy retention than with existing methods that remain exclusive to the centralised training paradigm.

## 3 BACKGROUND

This section describes the state-of-the-art sparse training method SWAT (Raihan & Aamodt, 2020); the way we adapt it to the FL contexts; and the related challenges that would need be addressed.

### 3.1 SPARSE WEIGHTS AND ACTIVATIONS TRAINING

The SWAT framework embodies two strategies in the training process. During each forward pass, the weights are partitioned into active weights and non-active weights by a `top-K` (in magnitude) operator and only the active weights are used. For the $l^{th}$ layer in the model, the layer maps the input activations $a_{l-1}$ onto feature maps $o_l$ using function $f_l$: $o_l = f_l(a_{l-1}, w_l)$. In this work we consider $f_l$ being the $3 \times 3$ convolution in the $l$-th layer. In the backward pass, the gradient of input activations ($\bigtriangledown a_{l-1}$) and the gradient of weights ($\bigtriangledown w_l$) are calculated represented by functions $G_l$ and $H_l$, as shown below:

$$\bigtriangledown a_{l-1} = G_l(\bigtriangledown a_l, w_l) \qquad (1)$$
$$\bigtriangledown w_l = H_l(\bigtriangledown a_l, a_{l-1}) \qquad (2)$$

Then in the backward pass, the retained layer inputs $a_{l-1}$ are also partitioned into active and non-active by using the same `top-K` procedure. This results in full gradients and active weights being used in Eq. 1, while full gradients and active activations are used in Eq. 2. It is worth noticing

that even weights and activations and sparsified in the forward and backward pass, the gradients generated through the training process are dense. Therefore, the resulting model is a dense. The compute cost of updating weights $w_l$ given a dense $\bigtriangledown w_l$ tensor is negligible compare to the savings due to performing the underlying convolutions in Eq.1&2, as this is essentially a weighted sum.

## 3.2 FROM CENTRALISED TO FEDERATED SPARSE TRAINING

A direct adaptation of the SWAT framework to the FL setting could be done by framing each local training stage on a client as an instance of centralised training. However, one major difference between centralized training and FL is the notion of *client statefulness*. In a centralised scenario, each example in the training set is seen multiple times, once per epoch, allowing for the model to converge to a stable distribution of weights. This scenario is more suitable for sparsification.

On the other hand, in a typical *cross-device* scenario, client's availability is low and new data points are continuously being presented to the system as new clients participate in training rounds. This means that clients are likely to participate only once. Such training behaviour inevitably leads to distributions of weights that change over time, making the application of sparsity inducing methods more difficult as a result.

## 4 SPARSE TRAINING FOR FEDERATED LEARNING

As a first step, we conduct preliminaries investigations and measure SWAT's performance when directly applied to FL without any adaptation. This section describes the experimental protocol (Section 4.1), the obtained baseline results (Section 4.2) and a sparsification effect analysis to highlight the weaknesses of this approach in Section 4.3.

## 4.1 EXPERIMENTAL SETUP

While SWAT is used across the experiments as the standard sparsification methodology, results also depend on various FL-specific hyper-parameters. Federated learning is simulated with the Virtual Client Engine (VCE) of the Flower toolkit (Beutel et al., 2020) enabling us to scale to a large number of clients within in a single machine. Datasets and hyper-parameters are detailed below.

**Datasets.** Experiments are conducted on two image classification tasks of different complexity both in terms of the number of samples and classes: FEMNIST (Caldas et al., 2018) and CIFAR10 (Krizhevsky et al., 2009). FEMNIST is constructed by partitioning the data of the Extended MNIST (Cohen et al., 2017) based on the writers of the digit-character. We also include the Speech Commands dataset (Warden, 2018), where the task is to classify 1-second long audio clips. Further details for these datasets can be found in the Appendix A.5.

**Data partitioning.** We follow the latent Dirichlet allocation (LDA) partition method (Reddi et al., 2021; Yurochkin et al., 2019; Hsu et al., 2019) ensuring that each client is allocated the same number of training samples. The level of heterogeneity is governed by the parameter $\alpha$. As $\alpha \rightarrow \infty$, partitions become more uniform (IID), and as $\alpha \rightarrow 0$, partitions tend to be more heterogeneous. Our experimental evaluation considers $\alpha = 1.0$ and $\alpha = 1000$.

**Model Architecture.** Following the convention for CIFAR-10, a ResNet-18 (He et al., 2016) architecture is instantiated in the client side and aggregated on the server. We also make use of ResNet-18 for SpeechCommands. For FEMNIST, we employ the much smaller CNN first proposed in (Caldas et al., 2018). Further details for these architectures can be found in Section A.6 in the Appendix. The models are trained with SGD, and all experiments imply one local epoch (*i.e.* client epoch). An exponential decay defined as $\eta_t = \eta_{start} \exp(\frac{t}{T} \log(\eta_{\text{start}}/\eta_{\text{end}}))$ with $\eta_{\text{start}}$, $\eta_{\text{end}}$ the starting and last learning rates respectively is applied at training time. $T$ represents the total number of FL rounds.

**Client partitioning.** Following previous works, we propose to compose a pool of 100 clients with 10 active clients training concurrently in a given round (McMahan et al., 2017). We do this for all experiments except for FEMNIST, which comes pre-partitioned into 3597 clients. For this dataset we consider the setting of sampling 35 clients per round as in Caldas et al. (2018).

**Sparsity Ratios.** This work considers accelerating the convolutions involved during forward and backward propagation following a `Top-K` sparsity inducing mechanism at the weight level. As a

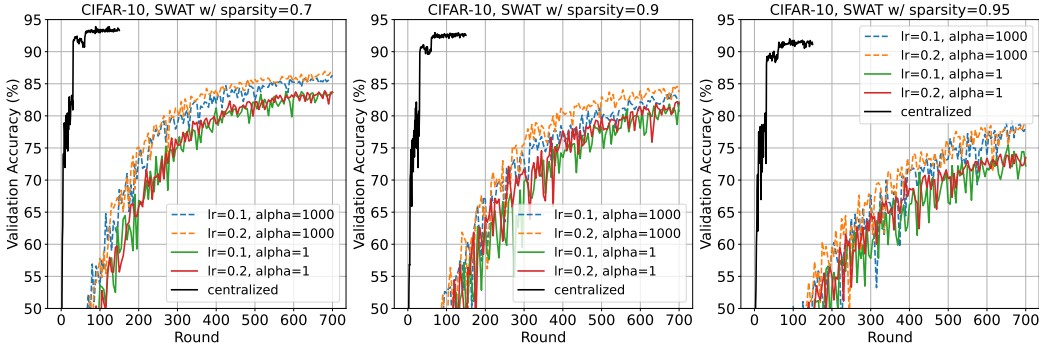

**Figure 1:** Comparison of validation accuracies in percentage of both centralised learning and FL on the CIFAR10 dataset with different sparsity and non-IID ratios. While centralised training suffers from minimal degradation at very high sparsity ratios (95%), the opposite happens for FL: we observe a 10% accuracy drop.

result, the expected sparse pattern would be unstructured, which can only be accelerated if tensors are sufficiently sparse. While *sufficient* is mostly hardware-specific, for the target platforms often considered in FL (e.g. mobile CPUs and GPUs) we set a minimum sparsity ratio ($sp$) of 90%, above which acceleration can be achieved (Wang, 2021). We include in our study $sp \in [0.7, 0.9, 0.95]$ in our initial evaluation. We expect minimal accuracy drop for both IID and non-IID at 0.7 sparsity.

### 4.2 BASELINES RESULTS

We begin by studying the effect of applying SWAT in both centralised learning and FL with the CIFAR10 dataset, and the results are given in Fig. 1. Applying SWAT in centralised training does not impact the validation accuracy much despite the high level of sparsity level, which is equivalent to the results from the original paper (Raihan & Aamodt, 2020) with a validation accuracy reaching 91.21% with a sparsity level of 95% against 93.32% for 70%.

We found sparse FL to be particularly sensitive to the learning rate and its scheduler. In particular, exponential decay annealing is crucial to reach relatively good performance. It is also clear from the curves that a higher learning rate of 0.2 reaches better accuracies in general than 0.1. As expected, however, FL offers lower levels of performance across all setups compared to centralised training.

In addition, and conversely to centralised training, plots show consistent drops in the validation accuracy with the increase of the sparsity level. It is worth noticing that the validation accuracy decreases by 4.60% and 2.78% while the sparsity level increases from 90% to 95% and 70% to 90% respectively for IID settings. Then the validation accuracy drops by 8.3% and 1.84% when sparsity levels increase from 90% to 95% and 70% to 90% respectively for non-IID settings. This highlights an important degradation of performance in non-IID settings with very high levels of sparsity.

### 4.3 SPARSIFICATION EFFECT ANALYSIS

As shown in Fig. 1, high levels of sparsity with FL (*e.g.* $\geq 90\%$) induce an significant drop of accuracy that is more noticeable than for centralised training. We aim at understanding this phenomenon to further reduce the gap between centralised and FL training by properly adapting sparsification. As a first step, we propose to investigate the behaviour of the neural weights from clients to clients under SWAT and FL with a sparsity ratio equals to 90%.

Indeed, an undocumented effect of SWAT occurs at inference time and may motivate an extension of the technique to work properly with FL. During training, SWAT partitions the weights in two sets: active and non-active. The former set is used as a sparsity map during any forward propagation (*i.e.* both at training and inference times). In fact, a part of the weights are simply dropped from the neural network. This implies that the neural network must remain sparsified when evaluating and inferring or the accuracy will drastically drops. For instance, a FL trained system on CIFAR10 that reaches an accuracy of 82% on the validation set will drop to 10% if the inference is done without sparsification.

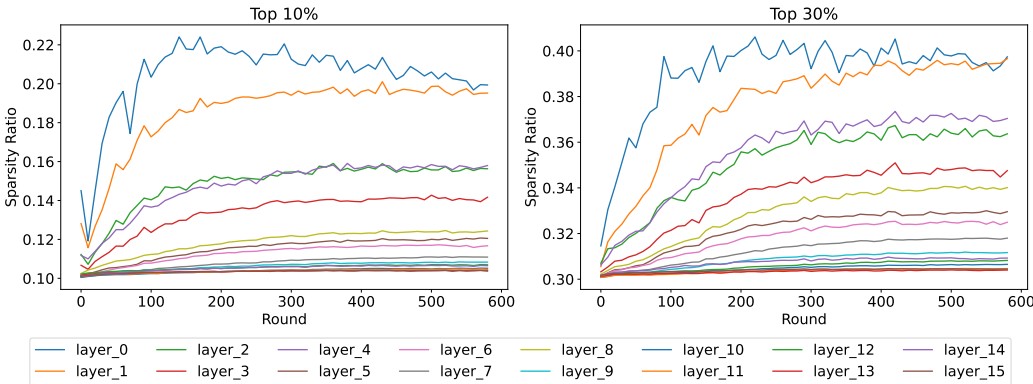

**Figure 2:** Evolution of the non-zero weights ratio after server aggregation (*i.e.* number of weights that are non-zero divided by the total number of parameter in that layer) of all CNN layers of a ResNet-18 trained on CIFAR10 with FL. Each of the 100 clients either send the `top-10%` or `top-30%` (*i.e.* weights with the highest norm) to the server.

It empirically validates that sparse training has an impact on the internal representation of a neural network making it dependent on the sparsification strategy during inference.

In practice, the latter behaviour is explained by the fact that, during centralised training, sparsified weights tend to be always the same *i.e.* some parameters are simply discarded. For FL, however, one may intuitively hypothesize that different clients will lead to different sparsification maps during the local training; preventing the creation of a global federated sparsification strategy. To investigate this, we propose to analyse the variations observed on the most active weights after aggregation on the server over the communication rounds.

Let us define the `top-K` weights as being the per-layer set of parameters with the highest norm. During each communication round, clients only send their `top-K` weights to the server while the remaining ones are set to zero. After aggregation on the server side, the resulting weight matrix informs us on the level of overlapping non-zero parameters observed on the clients. Indeed, every non-zero value obtained at a specific position will likely result in a non-zero value at the corresponding position in the aggregated weight matrix. For instance, if the number of non-sparse elements from the aggregated matrix is equivalent to the one of the clients ($K\%$) it means that all clients have the exact same `top-K` weights. With that in mind, we can define a non-zero ratio that is the number of non-zero parameters after aggregation divided by the the total number of elements in this layer. Thus, the higher this ratio is, the more different are the `top-K` weights sent from the clients. Fig. 2 depicts this ratio for different CNN layers for a non-IID CIFAR-10 with $K \in \{10, 30\}\%$.

First, a significant overlap exists between clients across all CNN layers. Indeed, the non-zero ratio almost never exceeds 0.40, meaning that at least 60% of the weights are not comprised in the `top-K` weights of the clients. This advocates for the fact that the most important weights for the current task tend to be the same across clients. Then, we can see that the non-zero ratio does not seem to be significantly impacted by changing $K$ from 10% to 30% as it only slightly increases for most CNN layers. This is explained by the fact that while clients may have different `top-10%` weights, they tend to have the same `top-30%` parameters: a specific non-zero weight that is reported as a `top-10%` element from a single client out of the selected ones is most likely to be reported as non-zero as well by more of them when we increase to `top-30%`. In short, `top-30%` will gather most of information about weights that are considered as being *important* for the task while keeping the same level of sparsity.

Second, we propose to examine the exact positions of the `top-10%` weights for certain CNN layers across a pool of 100 selected clients to better evaluate the overlap. Fig. 3 shows the weight matrix recorded every 20 communication rounds after aggregation on the server. Observations validate our intuition that most of zero and non-zero weights remain the same during the whole training.

Based on this analysis, we hypothesize that the degradation of performance observed with high levels of sparsity for FL is due to the dilution of important information during the aggregation process.

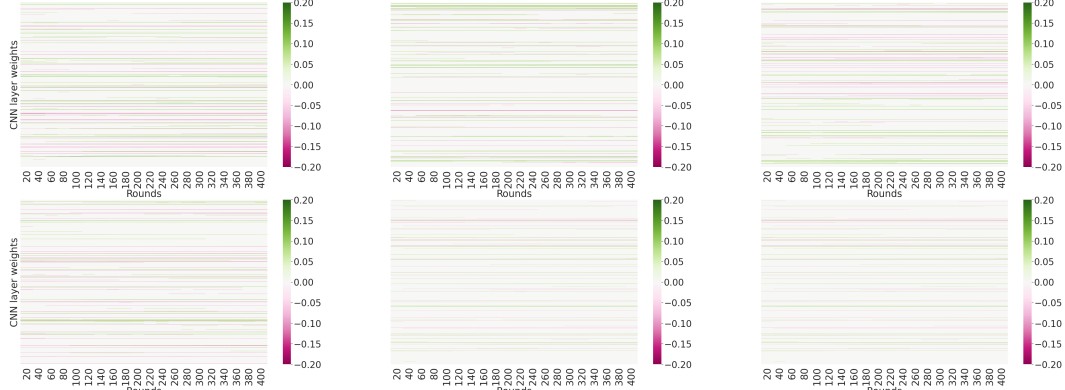

**Figure 3:** Heatmaps of 6 CNN layers (layer 4-9) in ResNet-18 when trained on CIFAR10 with 100 clients by only keeping the top 10% of weights. The weights are recorded every 20 communication rounds and *flatten* along the y-axis. The consistency across rounds (x-axis) indicates that, for the most part, the locations of non-zero weights remains constant. A larger version of this picture is given in the Appendix A.3

For instance, a weight that would be only sent by a single client as part of its `top-10%` parameters would not be *diluted* with the noise of all the others clients. Conversely, this very same weight may be completely corrupted if we send the entire dense model for aggregation.

## 5 ZEROFL: LOCAL SPARSIFICATION OF UPLINK COMMUNICATION

Motivated by our previous analysis which suggests that not all weights are necessary to be transferred to the central server for aggregation, we propose *ZeroFL*: a method that applies local sparsification before uplink communication. More precisely, we provide three strategies for local sparsification to improve the performance of sparse training while reducing the communication cost at the same time. By leveraging local sparsification, ZeroFL reduces the uplink communication footprint, hence reducing the noise aggregated on the central server observed in Section 4.3. In particular, if some updates are only sent by few clients, we force others clients to send zero in that particular positions of updates following our three strategies. After aggregation, the magnitude of these particular updates will be averaged with the uploaded values instead of being completely corrupted by the entire set of clients. All three methods are summarized in Algorithm 1.

**Top-K-Weights.** As shown in Sec. 4.3, only `top-K` active weights are involved in the validation and inference stages, and an important part of these weights tend not to change during training. Therefore, the first local sparsification method is to sparsify the `top-K` weights on the client-side before sending them back to the central server for aggregation. Then, and even though the positions of `top-K` weights largely overlap between each clients, only sending exactly a number of weights corresponding to the sparsity level might be too restrictive and prevents natural variations in the weights. Hence, we introduce a parameter denoted as the mask ratio $r_{mask}$, indicating the additional amount of weights that the selected clients will choose to send to the server after the local training.

Let $sp$ be the sparsity level of the local training. For instance, with $sp = 0.9$, and after the local training process, each selected client sparsifies their model by only keeping the top $(1 - sp + r_{mask})$ weights w.r.t their magnitude, while setting the rest to zero. In particular, if $sp = 0.9$ and $r_{mask} = 0.1$, the selected clients will send the `top-20%` to the central server for aggregation. The model produced after aggregation is not guaranteed to be sparse. If mask ratio equals to the sparsity level, the algorithm is degenerated to vanilla SWAT without local sparsification. However, and as shown in Fig.2, the resulting model is most likely to be sparse. Here, uplink communications are saved as only $(1 - sp + r_{mask})$ weights are sent as dense values.

**Diff on Top-K-Weights.** The idea behind this method is to send local weight-updates for the top-K weights, rather than sending the top-K weights themselves. Given $sp$ and $r_{mask}$, we first identify the weights that are the largest in magnitude by selecting the top $(1 - sp + r_{mask})$. The selected clients now send only the difference $d$ of these top $(1 - sp + r_{mask})$ weights with respect to the

---

**Algorithm 1** *ZeroFL*: Let us consider a cluster of $N$ total client with $n$ local data set and each with a learning rate $\eta_t$ at round $t$ with $T$ the total number of communication rounds. The client has the data set $n_k$. The number of local epoch is $E$ and the number of clients participating in each round is denoted as $K$. $w_t$ represent all the weights aggregated at round $t$ and $d_t$ the difference of weights.

---

**Central server does:**

    **for** $t = 0, ..., T - 1$ **do**

        Server randomly selects $K$ devices.

        **for all** $k$ in $K$ **do**

            Perform TrainLocally$(k, w_t)$

        **Aggregation**:

        **If Top-K-Weight then** $w_{t+1} \leftarrow \sum_{k=0}^{K} \frac{n_k}{n} w_{t+1}^k$

        **If Diff on Top-K-Weight then** $w_{t+1} \leftarrow w_t + \sum_{k=0}^{K} \frac{n_k}{n} d_{t+1}^k$

        **If Top-K Diff then** $w_{t+1} \leftarrow w_t + \sum_{k=0}^{K} \frac{n_k}{n} d_{t+1}^k$

**TrainLocally**$(k, w_t)$**:**

    **for** $e = 1, ..., E$ **do**

        Do local model training via SWAT with sparsity level $sp$.

        $w_e \leftarrow w_{e-1} - \eta_t \triangledown F(w_{e-1})$

    **Determine which weights to send for aggregation:**

        **If Top-K-Weight then return** top $1 - sp + r_{mask}$ weights.

        $d_{t+1}^k \leftarrow w_E - w_t$

        **If Diff on Top-K-Weight then return** $d_{t+1}^k$ of top $1 - sp + r_{mask}$ weights.

        **If Top-K-Weights Diff then return** top $1 - sp + r_{mask}$ of $d_{t+1}^k$.

---

originally received model $w_t$ as $d_{t+1} = w_E - w_t$ with $w_E$ the weights obtained after local training. The remaining differences are set to zero. In this way, after the aggregation in the central server, the weights that are not in the top $(1 - sp + r_{mask})$ part will remain the same as during the previous round, while only the top $(1 - sp + r_{mask})$ part of the weights will be updated.

**Top-K-Weights Diff.** Conversely to Diff on Top-K-Weights, this strategy proposed to first compute all the weights differences $d$ and then only send the top $(1 - sp + r_{mask})$ of them to the server. With this method, only highly moving weights will be considered.

All the aforementioned local sparsification methods lead to substantial reductions in uplink communication costs. More precisely, total communications will be reduced by a factor of $(r_{mask} - sp)/2$.

## 6 ZeroFL: Experimental Results

We conducted extensive experiments on CIFAR10, Speech Commands (Warden, 2018) and FEMNIST (Caldas et al., 2018) datasets. The CIFAR10 experiments follow the same experimental protocol as for baselines experiments. The three local sparsification strategies are compared with various mask ratio $r_{mask} = \{0.0, 0.1, 0.2\}$ and to vanilla SWAT without ZeroFL. Similarly to results obtained in Section 4.2, sparse training performs better with exponential learning rate decay. Hence, all setups are investigated with this scheduler. Table 1 reports the test accuracies achieved as well as the gain in communication cost when applying ZeroFL.

First, it is worth noticing that all three local sparsification methods with mask ratios higher than $0.1$ perform better or similarly to vanilla SWAT. The biggest improvement for $90\%$ sparsity is achieved with the *Top-K-Weights* method with a mask ratio of $0.2$, which increases the test accuracy by $0.4\%$. The largest improvement for $95\%$ sparsity is achieved with the *Top-K-Weights* method with a mask ratio of $0.2$, and the it increases the test accuracy by $1.5\%$.

For SpeechCommands, we reported the performance at communication round $200$ to show that ZeroFL achieves faster convergences and higher accuracies with mask ratio higher than $0$, especially for the non-IID setting. More results can be found in appendix A.4, which demonstrates higher performance at round $300$ when test accuracies are stabilized. With ZeroFL, the performance can be improved by $2.3\%$ for $90\%$ sparsity in the non-IID setting.

For FEMNIST we observe a similar trend when evaluating the different masking methods in ZeroFL: larger mask ratios result in better performing global models. However, sending the entire model from

**Table 1:** Results with ZeroFL on CIFAR10 and SpeechCommands for both IID ($\alpha$=1.0) and non-IID ($\alpha$=1000) settings. We report the test accuracy at 700, 200 and 1K communication rounds respectively for CIFAR10, Speech Commands and FEMNIST. We report the size (in MB) of the artifact to be transmitted to the server for aggregation, which has been compressed following the CSR sparse format representation. ZeroFL improves the performance while reducing the uplink communication cost up to a factor of $7.4\times$ compared to vanilla SWAT. For each sparsity level and dataset, we highlight in bold the best masking strategy. For clarity we do this on the non-IID results only. More results can be found in Appendix.

| Dataset | Sp Level | Mask Ratio | Full Model | | Top-K-W. | | Diff. Top-K-W. | | Top-K-W. Diff | | File Size | Comms. Savings |
|---|---|---|---|---|---|---|---|---|---|---|---|---|
| | | | IID | NIID | IID | NIID | IID | NIID | IID | NIID | | |
| CIFAR-10 (100 clients) | 90 % | — | 82.82±0.64 | 80.62±0.72 | | | | | | | 43.7 | 1× |
| | | 0.0 | | | 76.52±0.28 | 73.87±0.50 | 76.62±0.42 | 73.22±1.18 | 76.91±0.75 | 72.71±1.11 | 10.1 | 4.3× |
| | | 0.1 | | | 82.14±0.58 | 79.84±0.62 | 82.64±0.49 | 79.58±1.09 | 82.32±0.75 | 80.17±0.48 | 18.7 | 2.3× |
| | | 0.2 | | | 82.62±0.60 | **81.04±0.28** | 82.67±0.26 | 79.74±1.35 | 82.71±0.37 | 79.95±1.09 | 27.3 | 1.6× |
| | 95 % | — | 79.13±0.91 | 74.00±0.74 | | | | | | | 43.7 | 1× |
| | | 0.0 | | | 68.66±0.39 | 65.38±0.60 | 69.33±1.03 | 66.05±1.32 | 69.21±0.09 | 64.86±0.72 | 5.9 | 7.4× |
| | | 0.1 | | | 76.15±0.75 | 73.65±0.54 | 76.72±0.46 | 73.03±0.32 | 76.44±1.12 | 73.06±3.93 | 14.4 | 3.0× |
| | | 0.2 | | | 76.96±1.86 | **75.54±1.15** | 78.22±0.35 | 73.08±1.56 | 77.69±0.78 | 72.53±2.81 | 23.0 | 1.9× |
| Speech Commands (100 clients) | 90 % | — | 85.95±0.52 | 82.81±1.21 | | | | | | | 43.7 | 1× |
| | | 0.0 | | | 73.54±2.59 | 71.70±1.99 | 74.45±0.30 | 67.46±2.79 | 81.39±1.78 | 72.80±2.54 | 10.1 | 4.3× |
| | | 0.1 | | | 86.30±0.62 | 83.70±2.25 | 85.46±0.27 | 83.41±1.91 | 85.81±0.11 | 83.83±1.48 | 18.7 | 2.3× |
| | | 0.2 | | | 86.11±1.11 | 84.90±1.77 | 86.50±0.99 | **85.11±0.90** | 86.21±0.74 | 84.85±0.75 | 27.3 | 1.6× |
| | 95 % | — | 83.10±0.72 | 81.12±0.82 | | | | | | | 43.7 | 1× |
| | | 0.0 | | | 68.13±0.69 | 64.79±3.02 | 70.32±1.08 | 67.95±2.73 | 69.85±1.06 | 66.96±2.44 | 5.9 | 7.4× |
| | | 0.1 | | | 84.71±0.58 | **82.02±0.43** | 81.55±0.28 | 81.60±1.02 | 83.67±0.24 | 81.99±1.33 | 14.4 | 3.0× |
| | | 0.2 | | | 84.05±1.61 | 81.79±0.33 | 82.45±0.60 | 80.98±0.86 | 82.96±0.86 | 81.79±1.42 | 23.0 | 1.9× |
| FEMNIST (3597 clients) | 95 % | — | — | 83.34±0.41 | | | | | | | 23.0 | 1× |
| | | 0.0 | | | — | 76.79±0.90 | — | 77.16±2.07 | — | 77.02±0.93 | 1.3 | 17.7 × |
| | | 0.1 | | | — | 81.91±0.78 | — | 82.10±0.39 | — | 81.71±0.79 | 2.9 | 7.9 × |
| | | 0.2 | | | — | **83.78±0.19** | — | 83.01±0.27 | — | 82.54±0.65 | 4.4 | 5.2 × |

client to server seems to be key to obtain good results. We hypothesise this is because FEMNIST is a much more challenging dataset (62 classes) and the architecture used is relatively simple and less sensible to sparsification as a result.

Overall, performance improved with mask ratios between 0 and 0.2, indicating that there exist an optimal interval level. Indeed, a mask ratio equal to the sparsity level degenerates the system to the vanilla SWAT, which obtains worse results. It also implies that there is a trade-off between communication cost and performance. By using mask ratio of 0.1, each selected client performs local sparsification with an effective sparsity level of 80%, hence sending more data that for lower mask ratios values. In Section A.1 we densely evaluate the impact of $r_{mask} \in [0.1, 0.9]$ and, we observe no clear benefit of choosing larger ratios over smaller ones (e.g. 0.1). We briefly elaborate on this in the Appendix.

ZeroFL enables us to reduce the performance degradation observed with high levels of sparsity with FL while not completely alleviating it. During communication, each weight matrix from the model is vectorised and transmitted using the Compressed Sparse Row (CSR) (Tinney & Walker, 1967) format. Such representation requires exactly one integer index for each non-zero weight value in the model. Table 1 shows the level of compression and reduction in communication for different level of mask ratios. Compared to the original 43MB dense model, uplink communications are reduced by factors of $7.5\times$, $3.0\times$ and $1.9\times$ for mask ratios of 0.0, 0.1 and 0.2 respectively, with a sparsification ratio of 95%. Communication savings are calculated as the size ratio between original and compressed models; considering both weights and indices in the CSR file.

# 7 CONCLUSION

In this work, we consider the challenges of inducing high level of sparsity to accelerate on-device training for federated learning. We provide the first framework to leverage sparsity as a mechanism to accelerate on-device training without the server imposing any restrictions. We study on the unique perspective that arise when introducing sparsity at training time, hence motivating us to propose innovative state-of-the-art off-the-shelve sparsification methods to the FL domain *ZeroFL*. The method achieves +1.5% accuracy while reducing $1.9\times$ uplink communication costs compared with competitive baselines for CIFAR10 at 95% sparsity, and +2.3% accuracy for non-IID SpeechCommands at 90% sparsity. Our findings call for further investigations on the device-oriented optimisation of federated learning to motivate realistic deployments of this training methodology.

## ACKNOWLEDGEMENTS

This work was supported by the UK's Engineering and Physical Sciences Research Council (EPSRC) with grants EP/M50659X/1 and EP/S001530/1 (the MOA project) and the European Research Council (ERC) via the REDIAL project (Grant Agreement ID: 805194). Part of this work was performed using HPC/AI resources from GENCI-IDRIS (Grant 2021-A0111012991).

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

# A   APPENDIX

## A.1   IMPACT OF $r_{mask}$ IN GLOBAL PERFORMANCE

We hypothesised in Section 6 that increasing the masking ratio, $r_{mask}$, which is used to limit the amount of non-zero weights that would be uploaded to the central server after each client completes its local training, would yield better global model performance. However, as it can be observed in Figure 4, test accuracy does not monotonically increase with $r_{mask}$ and lower values (i.e. 0.1) that enable large savings in upload communication could perform as good as with larger values (e.g. 0.6) after fine-tuning.

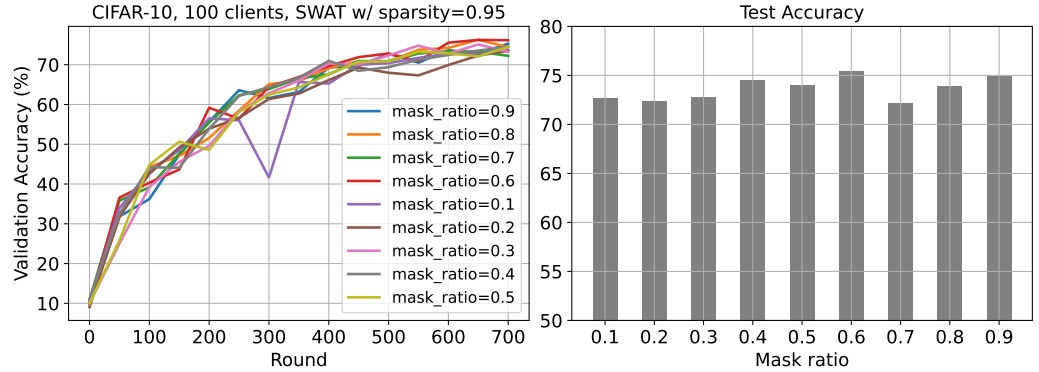

**Figure 4:** Larger masking ratios do not offer a clear advantage over smaller values (e.g. 0.1) despite them uploading a larger portion of the model parameters to the server after each round of local training. This experiment is conducted using a non-IID partitioning of CIFAR-10.

## A.2   CIFAR-10 WITH FEDADAM STRATEGY

Table 2 shows the performance of the proposed masking methods in ZeroFL when evaluating CIFAR-10 on the non-IID setting using the FedAdam (Reddi et al., 2021) aggregation strategy. The results are inline with what was first observed in Table 1 in Section 6: at large sparsity ratios (0.95) only ZeroFL making use of

**Table 2:** CIFAR-10 with FedAdam for the non-IID setting, 100 clients and using 10 clients per round.

| Sparsity Level | Mask Ratio | Full Model NIID | Top-K-W. NIID | Diff. Top-K-W. NIID | Top-K-W. Diff NIID | File Size | Comms. Savings |
|---|---|---|---|---|---|---|---|
| 90 % | — | 83.38 | | | | 43.7 | 1× |
| | 0.0 | | 83.22 | 82.14 | 83.43 | 10.1 | 4.3× |
| | 0.1 | | **84.01** | 81.58 | 83.60 | 18.7 | 2.3× |
| | 0.2 | | 83.67 | 83.29 | 82.79 | 27.3 | 1.6× |
| 95 % | — | 80.69 | | | | 43.7 | 1× |
| | 0.0 | | 81.11 | 80.67 | 80.45 | 5.9 | 7.4× |
| | 0.1 | | 81.02 | 80.29 | 80.09 | 14.4 | 3.0× |
| | 0.2 | | **83.30** | 81.35 | 81.25 | 23.0 | 1.9× |

## A.3   HEATMAP VISUALIZATIONS

As mentioned in Fig. 3, a bigger version of heatmap for one of the CNN layer (layer 4) is shown in Fig. 5 for better reference. Heatmap plots for SpeechCommands can be found in Fig. 6, which shows the similar results.

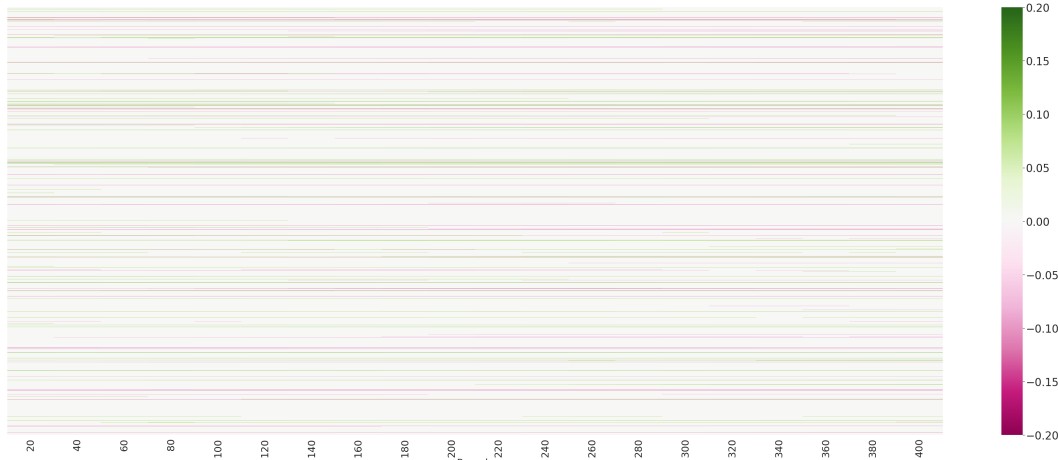

**Figure 5:** Heatmap, in bigger scale, of one CNN layers (layer 4) in ResNet-18 when trained on CIFAR10 with 100 clients by only keeping the top 10% of weights. The weights are recorded every 20 communication rounds and *flatten* along the y-axis. The consistency across rounds (x-axis) indicates that, for the most part, the locations of non-zero weights remains constant.

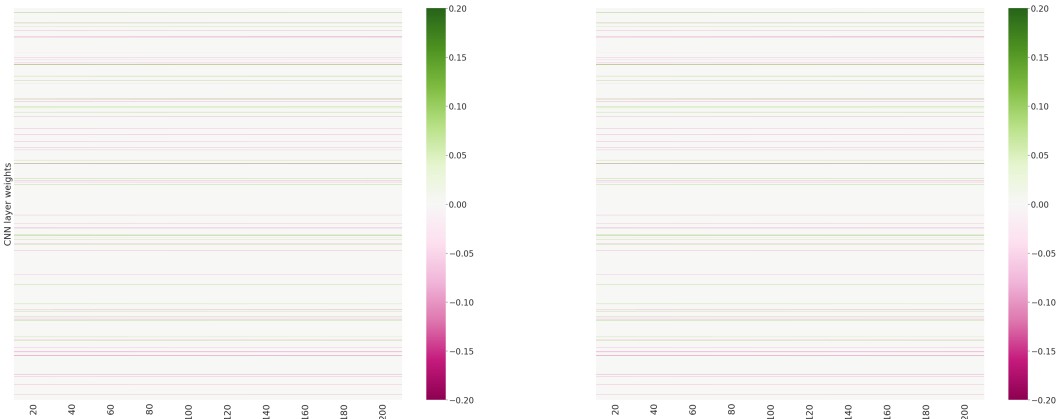

**Figure 6:** Heatmap, of two CNN layers (layer 4 and 9) in ResNet-18 when trained on SpeechCommands with 100 clients by only keeping the top 10% of weights. The weights are recorded every 20 communication rounds and *flatten* along the y-axis. The consistency across rounds (x-axis) indicates that, for the most part, the locations of non-zero weights remains constant.

## A.4 ADDITIONAL RESULTS FOR SPEECHCOMMANDS

Table 3 reports the performance of ZeroFL SpeechCommands after 300 communication rounds for both IID and non-IID settings. Combined with Table 1, the results show consistent better performance for non-IID settings for mask ratio higher than 0. The increase in accuracy at rounds 200 in Table 1 shows faster convergence of ZeroFL compared with baselines, while improvement in accuracies at rounds 300 in Table 3 demonstrates the better performance of ZeroFL overall.

## A.5 DATASETS

This work considers two image classification tasks of different complexity both in terms of the number of samples and classes: CIFAR10 (Krizhevsky et al., 2009) and FEMNIST (Caldas et al., 2018) with 10 classes and 62 classes respectively. The former is comprised of 60K 32×32 RGB images for training and 10K images for test. The latter, results in over 652K images for training and over 165K for test. FEMNIST images are 28×28 and grayscale. In both scenarios we randomly extract 10% out from the training set for validation. This is done at the client level, i.e., the validation

**Table 3:** Experimental results with ZeroFL on SpeechCommands using 100 clients. We evaluate both IID ($\alpha$=1.0) and non-IID ($\alpha$=1000) settings. The table reports the highest test accuracy within 300 rounds.

| Sp Level | Mask Ratio | Full Model | | Top-K-W. | | Diff. Top-K-W. | | Top-K-W. Diff | | File Size | Comms. Savings |
|---|---|---|---|---|---|---|---|---|---|---|---|
| | | IID | NIID | IID | NIID | IID | NIID | IID | NIID | | |
| 90 % | — | 87.47±1.50 | 85.36±1.21 | | | | | | | 43.7 | 1× |
| | 0.0 | | | 76.06±2.23 | 74.99±1.85 | 76.55±1.37 | 71.60±2.88 | 84.51±1.10 | 75.77±2.02 | - | 4.3× |
| | 0.1 | | | 87.63±0.42 | 85.44±1.46 | 86.60±0.31 | 85.97±1.31 | 87.40±0.41 | 85.03±1.82 | - | 2.3× |
| | 0.2 | | | 87.52±0.44 | 86.20±2.31 | 87.35±0.48 | 86.69±1.07 | 87.97±0.96 | 85.46±1.45 | - | 1.6× |
| 95 % | — | 85.17±1.16 | 83.21±1.88 | | | | | | | 43.7 | 1× |
| | 0.0 | | | 71.97±0.70 | 69.40±3.13 | 73.85±0.75 | 71.39±1.36 | 73.10±0.75 | 69.68±0.13 | - | 7.4× |
| | 0.1 | | | 86.42±0.64 | 84.52±1.37 | 83.91±0.77 | 83.73±0.60 | 84.40±0.98 | 84.06±2.44 | - | 3.0× |
| | 0.2 | | | 85.07±0.64 | 84.31±1.56 | 84.79±0.66 | 82.86±1.81 | 85.29±0.45 | 83.39±1.80 | - | 1.9× |

set for each client is extracted from each client's training partition. This is done to ensure that the validation set is representative of the underlying distribution.

In addition, we also perform analysis on the Speech Commands dataset (Warden, 2018) which consists of $65K$ 1-second long audio clips of 30 keywords, with each clip consisting of only one keyword. We train the model to classify the audio clips into one of the 10 keywords - "Yes", "No", "Up","Down", "Left", "Right", "On", "Off", "Stop", "Go", along with "silence" (*i.e.* no word spoken) and "unknown" word, representing the remaining 20 keywords from the dataset. The training set contains a total of $56,196$ clips with $32,550$ ($57\%$) samples from the "unknown" class and around 1800 samples ($3.3\%$) from each of the remaining classes, hence the dataset is naturally unbalanced. Similarly to (Zhang et al., 2018), each audio clip is preprocessed and $32{\times}32$ MFCC features are extracted and fed to the CNN backbone, a ResNet-18 as described in Section 4.1.

## A.6 MODELS AND HYPERPARAMETERS

For ResNet-18, all $3{\times}3$ convolutional layers implement sparse training as discussed in Sec. 3.2. As it is common with other optimizations, we leave the input layer unchanged (i.e. performing standard dense training). This architecture results in 11M parameters.

For FEMNIST, we borrow the much smaller CNN first proposed by Caldas et al. (2018). It is comprised of two $5{\times}5$ convolutional layers (each followed by a $2{\times}2$ maxpool laye) and two linear layers. We sparsify both convolutional layers and the first linear layer but leave the final layer unchanged (i.e. dense). This architecture results in parameters 6.6M parameters, which the first linear layer accounting for 97% of the memory footprint.

All our experiments make use of the following hyperparameters. The start and end learning rate are 0.1 and 0.01 respectively for CIFAR10, 0.01 and 0.001 for Speech Commands, and 0.004 and 0.001 for FEMNIST.

