# OpenReview forum: "ZeroFL: Efficient On-Device Training for  Federated Learning with Local Sparsity"
_ICLR.cc/2022/Conference — ICLR 2022 Poster_

### Official Review · Reviewer_Fw5w · 2021-10-29

**Correctness:** 4
**Technical Novelty And Significance:** 2
**Empirical Novelty And Significance:** 3
**Recommendation:** 5
**Confidence:** 4

**Main Review:**

This submission is reasonably well-written and easy to follow. The idea is simple to implement and seems to work, at least in this specific setting that the authors considered. This constitutes a nice advantage to ZeroFL. Nevertheless, I believe this submission needs quite a bit of work before I can recommend for acceptance:

- There are comparisons with other relevant baselines missing; e.g., federated dropout (which can be done on the weight level as well) is missing.
- The overall contribution seems incremental compared to the prior work of SWAT. Furthermore, the authors discuss how SWAT also employs activation sparsification but they seem to not use it at all in their experiments.
- The overall state of the experimental evaluation is a bit weak; there is only one dataset considered where the number of clients is relatively small (100). What happens on datasets with more users for example on EMNIST / FEMNIST? Furthermore, it seems that the authors show results from single seeds and therefore it is hard to see what is the stability of their results.
- The authors don’t talk about the extra cost of transmitting the indices of the non-zero values in the main text (although they do mention the CSC format in the table caption), which can be an important additional communication cost in the case of unstructured pruning. Furthermore, at section 5 at the end, the authors mention $0.5(r_{mask }- sp)$ savings and it is not clear to me how is that derived and whether it does take into account sending the indices of the non-zero values as well.

As for other general feedback:
- At the first bullet before the related work, the authors argue that with the way their sparsity is realised  the privacy is better preserved. Could they perhaps elaborate more? To me it seems that this can actually be worse for privacy as each user can have its own sparsity pattern and thus reveal additional information compared to a server imposed sparsity pattern which is the same for all users.
- This work seems to be geared towards upload compression, although to me it seems that download compression is also feasible. Have the authors experimented with such an idea?
- The algorithm needs some improvements, e.g., add an if statement for the different choices, as it can be confusing as it is now. Furthermore, the details Figure 3 are quite hard to see, so I would encourage the authors to use a different colour palette and intensity for the lines.
- At the caption of table 1 the authors mention a 7.4x improvement over vanilla SWAT but the 7.4x seems to be on the dense model, i.e., when SWAT is not employed. Could the authors elaborate on what they compare against?
- Sometimes the authors refer to top-K and sometimes to top-K%. It would be better if they pick one for consistency.


**Summary Of The Paper:**

This work proposes ZeroFL; a method that allows for sparse neural network training at the edge along with reduced upload communication, both of which being important aspects in federated learning.  ZeroFL essentially follows a prior work on sparse neural network training, namely SWAT, and adapts it appropriately for the federated setting. The SWAT method works by setting a target weight sparsity percentage (denoted by $sp$) which is enforced in both the forward and backward pass by a top-k operation on the weights. This allows for using sparse convolutions and thus offering training time speed-ups. The authors note that it is important to use the same set of sparse weights for inference as well, in order to not affect performance.

ZeroFL then applies this idea in the federated setting by letting each client perform SWAT type of training with a target sparsity of $sp$. Instead of the clients then communicating to the server the exact set of sparse weights obtained at the end of their local training procedure, ZeroFL proposes to add an extra “slack” $r_{mask}$ and communicate $1 - sp + r_{mask}$ percentage of weights (instead of $1 - sp$). The authors motivate this change via the results of an ablation study which showed significantly lower performance when directly applying SWAT to the federated setting. They argue that with the extra $r_{mask}$ weights, a “cleaner” training signal from the clients can be given as these weights are not “corrupted” by the weights of the other clients when averaging at the server. The authors further propose three separate weights of updating the server model, i.e., 1) using the client top-k weights, 2) using the difference between the server and client top-k weights and 3) using the top-k differences between the server and client weights.

The authors experimentally validate the ZeroFL method on cifar 10 using 100 clients and both iid and non-iid splits.


**Summary Of The Review:**

Overall, while ZeroFL is simple, it does seem to be a straightforward extension of SWAT to the federated setting with a very minor modification. The experimental evaluation is rather weak (experiments on a single dataset) and there is an absence of other critical baselines (such as federated dropout). These are the primary issues that lead me to recommending rejection.

---

> ### Author Response · Authors · 2021-11-23
> **Answers and changes for Reviewer 3 (1 / 3)**
>
> We are grateful to Reviewer 3 for this detailed review and finding that our manuscript is well-written, easy to follow and that is implement a simple idea that *seems to work seems to work, at least in this specific setting that the authors considered*. In the following, we hope that, by addressing every single concern and modifying the manuscript accordingly, we will be able to broaden the considered settings hence increase the validity of our research to Reviewer 3 and the readers.
>
>
> 1. _There are comparisons with other relevant baselines missing; e.g., federated dropout (which can be done on the weight level as well) is missing._ — We consider that a paper introducing the use of sparsity during training to accelerate forward and backward propagation significantly differs from other existing techniques which primarily rely on extracting sub-models from a large global model and train them in the standard fashion, i.e., using dense convolutions. Because our work is the first proposing sparsity at training time in the context of FL, we consider presenting an extended study instead of introducing comparison to other works that, although sharing a similar goal, are fundamentally different. We agree that a different type of work from the one here presented considering the range of techniques to accelerate FL training as a whole should include methods as the one suggested. These should include pruning-based methods where sub-models are derived and sent to models (e.g. as in FederatedDropout, FjORD, SplitMix) and others cited in the Related Work section. These techniques are orthogonal to the use of sparsity to accelerate computations, which is the object of our study.
>
>
>
> 2. _The overall contribution seems incremental compared to the prior work of SWAT. Furthermore, the authors discuss how SWAT also employs activation sparsification but they seem to not use it at all in their experiments._ — We argue that our contributions cannot be labelled as “incremental” primarily because what we present here is the first work that proposes the use of sparsity to accelerate training, study the unique phenomena that arise when doing so (Section 4.1) and propose a set of techniques that (1) substantially improve the performance of the global model from what SWAT alone can achieve (over +3% higher accuracy in some settings) and (2) enable a reduction in communication costs. This last point is not possible with vanilla SWAT since it does not learn a sparse model (instead it uses sparse operations to exclusively speedup convolutions and matrix multiplies during training). Other works (Federated Dropout, FjORD, others that rely on distillation upon aggregation) follow other approaches to make local training more lightweight, primarily in the form of pruning in both structured and unstructured setups. These methods are different from ZeroFL in the sense that clients train sub-models by making assumptions about the clients (e.g. their compute capability, amount of data, etc). Our work does not make any assumptions and as a result leaves up to the clients to “decide” which weights to treat as non-zero for each batch. With regards to the sparse activations, these are used to speedup the computation of the gradients w.r.t weights. But we acknowledge they are not utilised when assessing which weights to upload to the server. This could be an interesting line of future work.
>
>
> 3. _The overall state of the experimental evaluation is a bit weak; there is only one dataset considered where the number of clients is relatively small (100). What happens on datasets with more users for example on EMNIST / FEMNIST?_ — We share this concern with Reviewer 3 (and it is also shared by Reviewer 1 and 2). Therefore, and as requested, we added new experiments to support our claims and solve this issue. We would like to inform Reviewer 3 that not all the results are in the main paper (i.e. some are in the appendix) as this would simply not fit in the current state of the manuscript. Hence, we added two new datasets (FEMNIST from the LEAF FL benchmark, SpeechCommands) with one new modality (speech), one extended scenario for further analysis on the weight distribution and results of ZeroFL (IID CIFAR10) and one different strategy (FedAdam on CIFAR10). We also agree with Reviewer 3 that reporting results for a single seed may seem dangerous with respect to the claims. However, we would like to highlight that every single scenario already is extremely compute and energy intensive (more than 32 GPUs per batch of experiments for 24 hours). Nevertheless, we decided to agree, and completed Table 1 with results for CIFAR-10 and SpeechCommands for both IID and non-IID settings, and FEMNIST using non-IID (since this dataset is naturally partitioned). **In the Appendix we also included results for CIFAR-10 non-IID using FedAdam and observed that ZeroFL stills offers large gains at 95% sparsity level.**

---

> > ### Author Response · Authors · 2021-11-23
> > **Answers and changes for Reviewer 3 (2 / 3)**
> >
> >
> > 4. _The authors don’t talk about the extra cost of transmitting the indices of the non-zero values in the main text (although they do mention the CSR format in the table caption), which can be an important additional communication cost in the case of unstructured pruning. Furthermore, in section 5 at the end, the authors mention equation savings and it is not clear to me how is that derived and whether it does take into account sending the indices of the non-zero values as well._ — As it is unclear from the current version, we also changed this part in  Section 6 to During communications, each weight matrix from in the model vectorized and transmitted using an efficient sparse representation (CSR) (Virtanen et al., 2020).  Such representation requires exactly one integer index for each non-zero weight value in the model. Communication savings are calculated as the size ratio between original and compressed models; considering both weights and indices.**A similar comment is added to Table 1’s caption.**
> >
> >
> > 5. _At the first bullet before the related work, the authors argue that with the way their sparsity is realised the privacy is better preserved. Could they perhaps elaborate more? To me it seems that this can actually be worse for privacy as each user can have its own sparsity pattern and thus reveal additional information compared to a server imposed sparsity pattern which is the same for all users._ — Our intention was to transmit that with ZeroFL we are not compromising the level of privacy that FL can achieve “out-of-the-box” since the server still doesn’t access any data from the clients and, we believe that nothing that ZeroFL introduces (i.e. the different masking strategies) or the experimental protocol would lower the privacy. We acknowledge the way it was phrased might result in confusion and we have therefore remove it. However we still want to stress that the contributions of this work (i.e. the different masking methods to retain most accuracy at high sparsity ratios) do not require the server from knowing or accessing client-specific (and therefore private) information.
> >
> >
> > 6. _This work seems to be geared towards upload compression, although to me it seems that download compression is also feasible. Have the authors experimented with such an idea?_ — We explored this early in our project when we experimented with the different masking methods. However, we found that clients, even when operating at high sparsity ratios, still benefited significantly from receiving a dense (i.e. unsparsified) model from the server. We agree with the Reviewer that this would make the use of sparsity to accelerate training even more advantageous. We leave this as future work.
> >
> > 7. _The algorithm needs some improvements, e.g., add an if statement for the different choices, as it can be confusing as it is now. Furthermore, the details in Figure 3 are quite hard to see, so I would encourage the authors to use a different colour palette and intensity for the lines._ — Thank you for spotting these mistakes. **They are now fixed in the latest version.**
> >
> > 8. _At the caption of table 1 the authors mention a 7.4x improvement over vanilla SWAT but the 7.4x seems to be on the dense model, i.e., when SWAT is not employed. Could the authors elaborate on what they compare against?_ — Although SWAT makes use of sparsity to speedup training, it does not result in a sparse model. Our baseline is therefore the model resulting from applying vanilla SWAT to the FL setting (which results in a model as large as the standard ResNet18 in the case of CIFAR-10). One of the contributions in our work is to induce masking on the clients in order to sparsify the locally trained model and save on uplink communication costs. For this purpose we make use of the CSR format, which represents a matrix (or tensor) as two vectors: one with the non-zero values and, another with the positions of those values in the matrix. This therefore results in two equally long vectors, meaning that 50% of the resulting file contains indexes. There are other more sophisticated lossless storage formats that could be added on top of CSR to further compress it (for example `LZ78`) . Inducing some structuring on the locations of the non-zero weights would certainly help in achieving higher compression ratios.

---

> > > ### Author Response · Authors · 2021-11-23
> > > **Answers and changes for Reviewer 3 (3 / 3)**
> > >
> > > 8. _Sometimes the authors refer to top-K and sometimes to top-K%. It would be better if they pick one for consistency._ — Thanks again, this mistake is **now fixed**.
> > >
> > > We would like to thank again Reviewer 3 for the time invested in this detailed review. We hopefully have addressed every single concern leading to a definitely cleaner and more explicit version of this manuscript. We also hope that this will help Reviewer 3 to consider increasing the score of the submission.

---

> > > > ### Comment · Reviewer_Fw5w · 2021-11-24
> > > > **Response to rebuttal**
> > > >
> > > > I appreciate the effort that the authors put in the rebuttal and the new experimental results definitely improve this work. Some more comments from my side are
> > > >
> > > > - I disagree with the comment that this work is the first which considers sparsity during training in the FL setting. As previously mentioned "federated dropout" already did that, by letting the server apply a random mask to the network before communicating it to the client. In this way, it tackles directly the same problem as ZeroFL and, thus, I believe it has merit as a baseline in this work. This is also given the fact that federated dropout does download compression as well, whereas, as per the authors statements, download compression doesn't really play nicely with ZeroFL. Having said that, I do agree that ZeroFL is, theoretically, more flexible since it allows each client to determine its own sparsity pattern (whereas federated dropout fixes that in advance). Nevertheless, from Figure 3 it seems that in practice the clients don't really exploit this and the sparsity pattern is more or less the same among them. Does the sparsity pattern actually change throughout the local training at the client level or is it also more or less the same?
> > > >
> > > > - The colour palette of the figures 3,5,6 is still gray which, still, makes it hard to see.
> > > >
> > > > - I would encourage the authors to provide in future revisions some kind of standard deviation in the metrics of the table. Furthermore, reporting the highest accuracy throughout training is a bit misleading and I would encourage them to report the, average, accuracy at the end of training. This will at least make sure that the accuracy & communication savings reported are "aligned" at the table.
> > > >
> > > > I will increase my score accordingly.

---

> > > > > ### Author Response · Authors · 2021-11-26
> > > > > **Further experiments and answers (part 1)**
> > > > >
> > > > > Many thanks for Reviewer 3's review and comments. **Here, we will address the remaining concerns:**
> > > > >
> > > > > ​​1. **(part a)** *I disagree with the comment that this work is the first which considers sparsity during training in the FL setting. [...] Having said that, I do agree that ZeroFL is, theoretically, more flexible since it allows each client to determine its own sparsity pattern.*
> > > > >
> > > > > **Answer:**
> > > > > FederatedDropout stressed the importance of communication costs, local training costs and fairness (in the sense that no client should be left out due to their compute or memory constrained nature) in FL. We regard this paper as an important contribution to the field. However, we still argue that the differences between our work and FederatedDropout are substantially different especially if we compare exclusively how training takes place in each client: local training in FederatedDropout follows the standard training paradigm (i.e. a dense model, doing dense operations for forward and backward propagation) but it relies on a sub-model (a smaller version of the global model); on the other hand, ZeroFL, lets clients perform training leveraging sparse convolutions/matmuls during forward and backward propagation. These operations only translate into speedups for high enough sparsity ratios, generally above 90% if sparisty is unstructured (as it is the case in our work). Naturally, training with 90% (and even 95%) sparse weights and inputs poses some challenges during training. This training paradigm has only begun to be considered in purely centralized settings (e.g. the SWAT, ReSprop works) and we are the first to study its use in FL. As we stated in our previous rebuttal comment, what we propose is complementary to other techniques to accelerate local training (e.g. via sub-models as in FederatedDropout) and to save in uplink communications (as several works do).
> > > > >
> > > > > ​​1. **(part b)** *Does the sparsity pattern actually change throughout the local training at the client level or is it also more or less the same?*
> > > > >
> > > > > **Answer:**
> > > > > We ran a separate experiment (ResNet18 for CIFAR10 with sparsity 95% and alpha=1.0) to compare the overlap in the sparsity pattern induced to the weights and activation tensors across clients in a batch-wise fashion. As expected, we measured low overlap in the sparsity pattern induced to the activation tensors (while all sharing the same sparsity ratio). This was generally below 10% on average during the early stages of training and increased marginally for later rounds (never reaching over 15%). This is the expected result since clients have different data.
> > > > >
> > > > > On the other hand, the overlap on the sparsified weights tensors at every batch showed a much higher level of overlap (over 98%). Even though the overlap is very high, we still found that fixing it (which would make zeroFL a bit more similar to Federated Dropout and offer also reduced downlink communication costs) resulted in worse models, i.e. higher validation loss. We argue that the difference in sparsity pattern in the weighst (although highly overlapped across clients) is crucial to achieve good models at high sparsity ratios.
> > > > >
> > > > > ​​2. *The colour palette of the figures 3,5,6 is still gray which, still, makes it hard to see.*
> > > > >
> > > > > **Answer:** We agree with reviewer 3 that this Figure may be hard to read depending on the printing quality. This is why we added a larger version of the picture in the appendix. However, and as requested, we also decided to change the colour palette to a contrasted and color-blind friendly set (Orange — blue). Unfortunately, we cannot upload the image as we are not allowed to change the manuscript anymore …
> > > > >
> > > > > 3. *I would encourage the authors to provide in future revisions some kind of standard deviation in the metrics of the table. Furthermore, reporting the highest accuracy throughout training is a bit misleading and I would encourage them to report the average, accuracy at the end of training. This will at least make sure that the accuracy & communication savings reported are "aligned" at the table.*
> > > > >
> > > > > **Answer:** We totally understand that accuracy with standard deviation would provide additional metrics about the robustness and validity of the experimental results. Hence, please kindly find below tables reporting the validation accuracies averaged across 20 rounds at the end of training together with the standard deviation. More precisely, for CIFAR10, we reported the average validation accuracy from the rounds 680 to 700. For SpeechCommands, we reported the average validation accuracy from 280 to 300. For FEMNIST, we reported the average validation accuracies from 980 to 1000 rounds. As reviewer 3 may see from these tables, ZeroFL provides better validation accuracies compared with the baselines with smaller standard deviation, especially for the non-IID settings, which are more practical cases in the real world. These results will be added in an updated version of the manuscript.

---

> > > > > > ### Author Response · Authors · 2021-11-26
> > > > > > **Further experiments and answers (part 2)**
> > > > > >
> > > > > > ### CIFAR10 FedAvg results (Table 1):
> > > > > >
> > > > > > | Sparsity | Mask Ratio | Full Model| | Top-K Weight | | Diff Top-K Weight | | Top-K Diff | |
> > > > > > | ----- | ----- | ---- |-----|---- |--- |--- |--- |-- |--- |
> > > > > > |  |  |IID|NIID |IID|NIID |IID|NIID |IID|NIID |
> > > > > > | 0.9|-|83.96±0.78|80.71±1.11|||||||
> > > > > > | | 0.0|||76.03±0.30|72.62±0.77|76.83±0.31|72.43±0.91|77.61±0.52|74.76±1.15|
> > > > > > ||0.1| | | 82.72±0.43|81.16±0.27|83.34±0.79|80.36±0.52|82.24±0.28|80.02±0.72|
> > > > > > ||0.2|||83.04±0.24|80.91±0.68|82.43±0.45|81.53±0.32|83.23±0.37|81.44±0.08|
> > > > > > |0.95|-|78.28±0.50 |74.01±0.77|-|-|-|-|-|-|
> > > > > > ||0.0|||68.72±0.50|65.20±0.47|69.42±0.58|64.50±1.02|68.74±0.35|65.25±0.48|
> > > > > > ||0.1|||75.45±1.35|74.12±1.30|76.88±0.94|74.17±1.06|77.33±0.62|74.26±0.70|
> > > > > > ||0.2|||76.03±0.75|76.13±0.93|76.15±1.02|74.81±0.98|78.07±0.67|75.55±1.20
> > > > > >
> > > > > > ### CIFAR10 FedAdam results (Table 2):
> > > > > >
> > > > > > | Sparsity | Mask Ratio | Full Model | Top-K Weight  | Diff Top-K Weight |  Top-K Diff |
> > > > > > | ----- | ----- | ---- |-----|---- |--- |
> > > > > > |  | |NIID |NIID |NIID |NIID |
> > > > > > |0.9|-|77.86±2.98||||
> > > > > > ||0.0|-|83.16±0.62|82.94±0.31|83.42±0.85|
> > > > > > ||0.1|-|83.16±0.47|82.68±0.98|84.06±0.53|
> > > > > > ||0.2|-|81.91±2.03|82.55±0.50|82.38±0.94|
> > > > > > |0.95|-|-|76.62±1.08|||
> > > > > > ||0.0|-|80.88±0.62|80.94±0.69|80.89±0.31|
> > > > > > ||0.1|-|80.76±0.70|79.17±0.70|81.12±0.84|
> > > > > > ||0.2|-|82.71±0.97|79.75±1.14|80.11±1.16|
> > > > > >
> > > > > > ### Speech Commands results (Table 3):
> > > > > > | Sparsity | Mask Ratio | Full Model| | Top-K Weight | | Diff Top-K Weight | | Top-K Diff | |
> > > > > > | ----- | ----- | ---- |-----|---- |--- |--- |--- |-- |--- |
> > > > > > |  |  |IID|NIID |IID|NIID |IID|NIID |IID|NIID |
> > > > > > 0.9|-|89.02±0.19|84.56±1.00|||||||
> > > > > > ||0.0|||73.79±0.35|76.78±0.27|77.58±0.68|71.76±0.73|78.22±0.28|78.04±0.57|
> > > > > > ||0.1|||86.95±1.42|87.41±0.29|86.15±0.27|86.99±0.31|87.75±0.54|87.70±0.40|
> > > > > > ||0.2|||87.978±0.50|88.28±0.54|86.65±0.65|86.94±0.63|87.70±0.34|86.91±0.55|
> > > > > > 0.95|-|86.72±1.25|84.47±0.89|||||||
> > > > > > ||0.0|||71.74±0.44|71.6±0.45|73.90±0.45|72.37±0.80|73.61±0.65|72.22±0.21|
> > > > > > ||0.1|||86.25±0.66|85.07±0.54|83.34±0.46|83.93±0.72|84.87±0.30|85.79±0.50|
> > > > > > ||0.2|||84.74±0.62|85.58±0.78|84.01±0.48|84.17±0.92|85.76±0.59|85.27±0.49|
> > > > > >
> > > > > > ### FEMNIST results:
> > > > > >
> > > > > > | Sparsity | Mask Ratio | Full Model| Top-K Weight  | Diff Top-K Weight | Top-K Diff |
> > > > > > | ----- | ----- | ---- |-----|---- |--- |
> > > > > > |  |  |NIID|NIID |NIID|NIID |
> > > > > > 0.95|-|80.04±0.49||||
> > > > > > ||0.0||75.47±0.13|74.55±0.19|74.18±0.20|
> > > > > > ||0.1||77.93±0.13|76.82±0.38|74.18±0.20|
> > > > > > ||0.2||77.89±0.19|77.92±0.14|77.73±0.21|
> > > > > >
> > > > > > Again, it is great to be able to engage in a discussion with reviewers and we really hope that all the changes will cover the doubts and concerns raised. If not, we are looking forward to answering / addressing the remaining issues.

---

### Official Review · Reviewer_sb7x · 2021-11-02

**Correctness:** 4
**Technical Novelty And Significance:** 3
**Empirical Novelty And Significance:** 3
**Recommendation:** 6
**Confidence:** 4

**Main Review:**

The paper has following strengths:
1.	Indeed, for more realistic deployment and to move federated learning towards practical systems, it is important to accelerate the on-device training (and not just the inference which is mainly the current practice). So, the problem the authors have considered is important.
2.	The proposed strategies clearly indicate the improvements over existing centralized sparse training technique like SWAT.
3.	Some of the insights shared (like the non-sparse weights staying the same throughout FL) were interesting.

The paper has following weaknesses:
1.	The main weakness of this paper is the experiments section. The results are presented only on CIFAR-10 dataset and do not consider many other datasets from Federated learning benchmarks (e.g., LEAF https://leaf.cmu.edu/). The authors should see relevant works like (FedProx https://arxiv.org/abs/1812.06127) and (FedMAX, https://arxiv.org/abs/2004.03657 (ECML 2020)) for details on different datasets and model types. If the experimental evaluation was comprehensive enough, this would be a very good paper (given the interesting and important problem it addresses).
2.	One other thing (although this is not the main focus of this paper), the authors should provide comparisons between strategies that result in fast convergence (without sparsity) vs. sparse methods? For example, do non-sparse, fast convergence methods (like FedProx, FedMAX, and others) result in small enough number of epochs compared to sparse methods? Can the fast convergence methods be augmented with sparisity ideas successfully without resulting in significant loss of accuracy? Some discussion and possibly performance numbers are needed here.


**Summary Of The Paper:**

The paper investigates sparse training to accelerate on-device federated learning. It begins by taking an off-the-shelf sparse training method (SWAT) and analyzes its limitations. Then, the method proposes three strategies to improve upon SWAT. Experiments are conducted on CIFAR-10 dataset and show some improvements compared to SWAT.

**Summary Of The Review:**

Even though the reviewer is quite happy about the problem addressed in this paper, it lacks a solid experimental section. I would be willing to increase the score if the authors can present a more comprehensive results section.

Post rebuttal:
I have read the author response and other reviews. Thanks to the authors for a detailed rebuttal and new experiments. It addressed my concerns and so I have increased the rating from 5 to 6.

---

> ### Author Response · Authors · 2021-11-23
> **Answers and changes for Reviewer 2**
>
> The authors thank Reviewer 2 for considering that the problem we consider is important for FL, that our proposed approach obtains promising and better results and that some of our findings are interesting. Positive comments are very much appreciated! Now, we would like to carefully address every concern raised by Reviewer 2 and change the manuscript accordingly:
>
> 1. _The main weakness of this paper is the experiments section. The results are presented only on CIFAR-10 dataset and do not consider many other datasets from Federated learning benchmarks (e.g., LEAF). The authors should see relevant works like FedProx and FedMAX for details on different datasets and model types. If the experimental evaluation was comprehensive enough, this would be a very good paper (given the interesting and important problem it addresses)._  — We understand and agree with Reviewer 2 that complying with standards is necessary to ensure a standardized evaluation of new methods. Hence, and as proposed, we looked into LEAF and the FedProx paper, and we decided to add a dataset contained in both: FEMNIST. We also propose to add another modality following the request of Reviewer 1 with SpeechCommands (Speech). For FEMNIST, we also decided to sparsify MLP as well to further extend the experiments. New results suggest that our claims and findings extend well to other domains, datasets and architectures as they remain equivalent throughout the entire benchmark. We made changes to the manuscript according to those new results and to the remark of Reviewer 2. **In particular we substantially expanded Table 1 and added further results to the appendix including FedAdam strategy for CIFAR-10 in the non-IID setting and longer runs for the SpeechCommands dataset.**
>
>
> 2. _One other thing (although this is not the main focus of this paper), the authors should provide comparisons between strategies that result in fast convergence (without sparsity) vs. sparse methods? For example, do non-sparse, fast convergence methods (like FedProx, FedMAX, and others) result in small enough number of epochs compared to sparse methods? Can the fast convergence methods be augmented with sparsity ideas successfully without resulting in significant loss of accuracy? Some discussion and possibly performance numbers are needed here._  — As clearly stated by Reviewer 2, other (more contemporary) FL strategies may completely change the findings observed with FedAVG. To address this concern, we propose to validate our approach with FedAdam as well on the CIFAR10 setup. As the narrative of the results remain the same compared to FedAVG, we decided to not extend FedAdam to the other datasets and setups due to the important compute and energy cost of those numerous experiments (more than 32 GPUs are needed per batch of experiment). Indeed, FedAdam indeed improves the results over FedAvg, but still suffers from the same accuracy drop when sparsified and isn’t significantly faster to converge than FedAvg (i.e. to justify a non-sparse training). Changes made to the manuscript are: **We included in the Appendix (Table 2) results for CIFAR-10 with non-IID partitioning using FedAdam as strategy**. While at 90% sparsity, the our masking strategies can recover ~0.7% accuracy, for the much more challenging 95% sparsity setting, our method  shows 2.6% better accuracy retention compared to vanilla SWAT adapted to the FL setting.
>
>
> 3. _Even though the reviewer is quite happy about the problem addressed in this paper, it lacks a solid experimental section. I would be willing to increase the score if the authors can present a more comprehensive results section._ — As a summary, and even though all the new results are not in the main body of the paper, we added: two new datasets (FEMNIST from the LEAF FL benchmark, SpeechCommands) with one new modality (speech), one extended scenario for further analysis (IID CIFAR10) and one different strategy (FedAdam on CIFAR10).
>
>
> We would like to thank Reviewer 2 for both the positive comments, and the remarks and concerns that definitely led to an improved version of the submission. As we, at least from our perspective, addressed every comment of Reviewer 2, we hope that an increase of the score will be considered as well.

---

> > ### Comment · Reviewer_sb7x · 2021-11-29
> > **Addressed my concerns**
> >
> > I have read the author response and other reviews. Thanks to the authors for a detailed rebuttal and new experiments. It addressed my concerns and so I have increased the rating.

---

### Official Review · Reviewer_1bwd · 2021-11-03

**Correctness:** 2
**Technical Novelty And Significance:** 2
**Empirical Novelty And Significance:** 3
**Recommendation:** 6
**Confidence:** 3

**Main Review:**

Strengths:

1. The authors introduce a new problem of adding sparse local training to federated learning. The problem appears well motivated (sparse training can lead to significant power savings at the edge) and challenging (directly adding sparse training to conventional aggregation methods like FedAvg leads to significant accuracy drops).

2. The proposed solutions are good initial approaches backed by the empirical study and do appear to lead to some improvement in accuracy over the naive approach.

Weaknesses:

1. For a paper that relies entirely on empirical evidence for its claims, the empirical study appears a bit limited as it considers only a single dataset (CIFAR10) and a single model architecture (ResNet18). Without seeing results for some other image/text datasets and other model architectures (MLP/LSTM) it is difficult to completely believe the claims especially since the proposed solutions rely heavily on the claims that the positions of the top-K weights do not vary significantly across models and across rounds.

2. The experimental results are also a bit limited. In particular:

a) It is misleading to highlight both max accuracy and max comms savings values since the two correspond to different settings and thus cannot be achieved simultaneously. I would suggest removing the bold font for the max comms savings model and just reporting the comms savings for the different mask ratios. An interesting future direction would be to think of a metric that can combine the two since it is currently unclear if the communication saving (and local computation cost reduction) is significant enough to compensate for the drop in accuracy w.r.t conventional FL (black line in Fig. 1)

b) Since the authors acknowledge that accuracy is expected to first increase and then decrease on increasing mask ratio, it would be worth including results for all mask ratios in the appendix at least to see the trend clearly.

c) The results with i.i.d data partitioning are not presented. Since from Fig 1 it seems like accuracy drops even for the i.i.d case, it is important to show if the proposed schemes alleviate the issue in that case as well.

d) Lastly, following point 1, in my opinion all experiments should be performed with some other datasets and model architectures before the paper can be considered for acceptance.

3. (Minor) There doesn't seem to be any real difference between Top-K-Weights and the Diff on Top-K-weights in that the weight updates will be exactly the same in both cases as far as I can tell. If that is so then there are essentially 2 (and not 3) new schemes being proposed.







**Summary Of The Paper:**

The paper studies the problem of adding sparsity to local training in federated learning. Under the approach each node uses only the top-K weights for computations in the forward and backward pass of training (and only top-K activations for computations in the backward pass). The authors empirically show that adding sparsity in this way, while speeding up local training, significantly degrades the accuracy when directly used in federated learning. To mitigate this issue the authors propose to combine sparse training with sparse aggregation since their empirical study suggests that the accuracy drop may be due to dilution of weights that are active in only a few nodes. Experiments on CIFAR10 with non-IID splits using ResNet-18 as the model show some gain in accuracy along with a drop in communication cost when using the proposed schemes.

**Summary Of The Review:**

The authors introduce an interesting and important problem but given that the paper relies purely on empirical evidence for its claims the current evaluation is quite limited and not enough for acceptance. I find the claims fairly intuitive, however, and so would be willing to increase my score if more empirical evidence (in line with my comments above) is presented to back them.

Comments after rebutal : I appreciate the added experiments which certainly demonstrate that the proposed schemes are able to improve performance across different datasets. I have increased my score to reflect the same.

---

> ### Author Response · Authors · 2021-11-23
> **Answers and changes for Reviewer 1 (1/2)**
>
> First, we would like to thank Reviewer 1 for finding our work well-motivated and challenging. In the following, we will carefully address each concern raised by Reviewer 1 and report the changes done to the manuscript:
>
> 1. *”For a paper that relies entirely on empirical evidence for its claims, the empirical study appears a bit limited as it considers only a single dataset (CIFAR10) and a single model architecture (ResNet18). Without seeing results for some other image/text datasets and other model architectures (MLP/LSTM) it is difficult to completely believe the claims"* — We agree with reviewer 1, and we prepared a **set of new results to entirely address this**. In particular, and as proposed, **we investigated two other datasets** involving one different modality: **FEMNIST (Image), and Speech Commands (Speech)**. Please note that we introduced FEMNIST as it is part of the standard FL LEAF benchmark. To address the second issue related to the model, we investigated another architecture containing MLP on FEMNIST. Results can be found at the bottom of Table 1. In practice, all results are in line with our original findings, and we hope that this will enable Reviewer 1 to believe in our claims.
>
>
> 2. *I would suggest removing the bold font for the max comms savings model and just reporting the comms savings for the different mask ratios.*  —To improve the readability and as suggested, we removed the bold font on the max communication-saving model and only highlight it for non-IID results.
>
>
> 3. *"An interesting future direction would be to think of a metric that can combine the two since it is currently unclear if the communication saving (and local computation cost reduction) is significant enough to compensate for the drop in accuracy w.r.t conventional FL (black line in Fig. 1)."* — In line with Reviewer 1, we think that this tradeoff between communication-saving and the drop in accuracy is an open question. In fact, it is most likely that this will remain scenario-dependent as long as such a drop is observed (i.e. will not be a question anymore once we are able to reach the same level of accuracy). Indeed, in practice, some devices will simply not be able to operate in FL without a sparsification solution comparable to what we propose as their compute power would drastically hinder the user experience when used to train a model for a long time period (e.g. if training while charging / at night is unavailable).
>
>
> 4. *”Since the authors acknowledge that accuracy is expected to first increase and then decrease on increasing mask ratio, it would be worth including results for all mask ratios in the appendix at least to see the trend clearly.”* — We would like to apologize for the slightly misleading statement. In our experiments, the amount of masking did not have a significant impact on the final global performance, the accuracy and the observed drops are close to uniform, suggesting that even with very small masking ratios (which result in large communication savings) we can still achieve high accuracy after appropriate hyperparameter adjustment. **However, and as requested by reviewer 1, we added a Figure to illustrate this to the appendix and updated the text in Section 6**.
>
>
> 5. *”The results with i.i.d data partitioning are not presented. Since from Fig 1 it seems like accuracy drops even for the i.i.d case, it is important to show if the proposed schemes alleviate the issue in that case as well.”*  — We agree with Reviewer 1 that completely removing the IID scenario might look suspect. During the writing, we decided to discard those results as they were simply displaying the same behavior while decreasing the readability of the paper. Nevertheless, and as proposed by Reviewer 1, **we added those results back to Table 1 so readers can get a complete overview of the scenarios for CIFAR and SpeechCommands as well!**.
>
>
> 6. *”Lastly, following point 1, in my opinion all experiments should be performed with some other datasets and model architectures before the paper can be considered for acceptance.”. — **As answered previously (see point 1) we added two new datasets and one new model architecture.**
>
>
> 7. *”(Minor) There doesn't seem to be any real difference between Top-K-Weights and the Diff on Top-K-weights in that the weight updates will be exactly the same in both cases as far as I can tell. If that is so then there are essentially 2 (and not 3) new schemes being proposed."* — Let us please clarify this point: In short, Top-K-Weights will allow the weights with the largest magnitude to be sent while Diff will send the largest difference between the model received at the beginning of a round and the model obtained after local training. Weights with the largest magnitude are not necessarily the ones that are changing the most during local training between one round to another. This does not really impact the local training but impacts the aggregation.

---

> > ### Author Response · Authors · 2021-11-23
> > **Answers and changes for Reviewer 1 (2/2)**
> >
> > **Conclusion: Again, the authors would like to thank Reviewer 1 for the nice comments, good critics and remarks, and time spent doing this review. We carefully addressed all the raised concerns, and we hope that this will motivate Reviewer 1 to increase the score.**

---

> > > ### Comment · Reviewer_1bwd · 2021-11-27
> > > **Response to rebuttal**
> > >
> > > I appreciate the added experiments which certainly demonstrate that the proposed schemes are able to improve performance across different datasets. I have increased my score to reflect the same. While there is still room for improvement I think these results are okay for an initial work. In my opinion future work should look into designing more sophisticated aggregations schemes that can approach the performance of centralized SWAT and are also consistent across models (currently the best performing aggregation method is different for different settings which can make it difficult to decide which scheme should be used for a new dataset). I also have a couple of (minor) questions which are more for clarification and won't really affect my score:
> > >
> > > 1. What is the architecture of the model for the Speech Commands dataset? I could not find it mentioned in the paper.
> > >
> > > 2. In Fig 2 how is the non-zero weight ratio less than 0.3 when the clients send 30% of their weights (in the left part of the right plot the fraction of non-zero weights for some layers in the initial rounds is 0.26 etc) ? Is it because the weights cancel each other thus leading to zeroing out of some weight?

---

> > > > ### Author Response · Authors · 2021-11-30
> > > > **Answers to Reviewer 1**
> > > >
> > > > 1.  For SpeechCommands we used ResNet18 and keep the same 32x32 input resolution as with CIFAR10. For this speech dataset we transformed the 1s-long audio clips into 32x32 MFCC features (a common practice for this dataset). We included this in the Model Architecture paragraph in Section 4.1 and in Appendix A.5. We forgot to highlight in red the lines in section 4.1
> > > >
> > > >
> > > > 2.  As for the explanation on why the non-zero ratios shown in Figure 2 (right) go below 0.3 when clients send their top-30% weights, we couldn't find an explanation so decided to **re-run the experiment with the latest codebase** (the same we utilized to run all the additional experiments added during the rebuttal process). We now observe that no line goes below the 0.3 value in the y-axis. This is the expected result and we will update the pdf when possible. Additionally we ran the same experiment with when mask_ratio is set to 0.0 and to 0.1 and we observed the expected results: no layer in the global model had a non-zero ratio below 0.1 and 0.2 respectively. These experiments were run following the same protocol as those for Figure 2: CIFAR-10 non-IID (alpha=0.1) and sparsity ratio set to 90%.

---

> > > > > ### Comment · Reviewer_1bwd · 2021-11-30
> > > > > **Re**
> > > > >
> > > > > Thank you for the response

---

### Public Comment · ~Chaoyang_He1 · 2021-11-09
**Another line of work is not discussed**

Dear Authors,

For efficient training at the edge, we've published a paper at NeurIPS 2020. It's highly appreciated if you could compare/discuss with this line of work. Thanks.

Group Knowledge Transfer: Federated Learning of Large CNNs at the Edge. NeurIPS 2020
https://arxiv.org/abs/2007.14513

To address the resource-constrained reality of edge devices, we reformulate FL as a group knowledge transfer training algorithm, called FedGKT. FedGKT designs a variant of the alternating minimization approach to train small CNNs on edge nodes and periodically transfer their knowledge by knowledge distillation to a large server-side CNN. FedGKT consolidates several advantages into a single framework: reduced demand for edge computation, lower communication bandwidth for large CNNs, and asynchronous training, all while maintaining model accuracy comparable to FedAvg.

---

> ### Author Response · Authors · 2021-11-23
> **Very interesting work**
>
> Thank you very much for pointing out your work. It definitely is relevant to the overall problem: "efficient FL", hence we decided to cite it in our manuscript. However, we can not really compare as both works are really different in the way we approach the issue. Your work is a complete framework redesigning the way of doing FL. Our is an ad-hoc solution that can be applied to any existing FL strategy with minimal changes (i.e. alongside with a study specifically aiming at sparsity in the context of FL and nothing else)

---

### Author Response · Authors · 2021-11-23
**Summary and general answer to the reviewers**

We would like to start by thanking all the reviewers for the efforts they have put into examining our work and spotting its strengths and weaknesses. We genuinely think that all their comments helped us to significantly improve our submission by making the main ideas clearer and improving the justifications defending the proposed method and our claims. With the answers given bellow to each reviewer, **we took the time to systematically address each and every question or concern raised by all the reviewers, performing numerous costly additional experiments (hence the unfortunate delay in the response).  All changes are marked in RED in the updated manuscript. We do hope that the updated version will dissipate the doubts concerning certain parts of the method and encourage the reviewers to update their scores accordingly if satisfied.**

---

### Author Response · Authors · 2021-11-29
**Final summary**

As the discussion period ends, we would really like to thank the reviewers for their availability during this review process and for increasing their scores from 5,5,3 to 6,6,5. It is a pleasure for us to see that all the hard work added during the rebuttal period helped to create a much more appreciated and complete version of the manuscript. We have presented the first study on how the use of sparsity to accelerate common operators (e.g. convolutions and matrix-matrix multiplications) of DL models during training impacts model convergence and quality in FL scenarios. We think this work sets the foundations for future research aiming at accelerating FL workloads making use of even more complex model architectures, data distribution settings and, running on even more constrained devices. We now hope that this will be sufficient to convince everyone towards acceptance, allowing us and others to build new ideas on top of this.

---

### Decision · Program_Chairs · 2022-01-20

**Decision:**

Accept (Poster)

**Comment:**

This paper considers the problem of on-device training for federated learning. This is an important problem since, in real-world settings, the clients have limited compute and memory, and local training needs to be efficient. The paper shows that the standard sparsity based speed-up techniques that consider top-K weights/activations during forward and/or backward pass do not work well in the federated setting and proposes several solutions to mitigate this issue. The proposed solutions are demonstrated to work well on several datasets.

In their initial assessment, given that this is largely an empirical insights driven paper, the reviewers mainly expressed concerns about the experimental evaluation (e.g., only one dataset CIFAR10 and one architecture ResNet18) and lack of more baselines (e.g., Federated Dropout). The authors responded in detail to the reviews and also conducted additional experiments and the reviewers and authors engaged in discussion. As the discussion converged, the reviewers agreed that the revised manuscript addresses their key concerns and their assessment, on an average, are now learning largely towards a borderline accept.

I also read the reviews, the discussion, and read the paper. I think the paper is a good initial attempt at providing a general approach to enable on-device federated learning when the clients are lightweight devices (e.g. edge devices). Even though the study is somewhat preliminary, the current manuscript, after the revision during the discussion phase, is significantly improved version of the original submission and does address the key concerns from the reviewers. Overall, I would rate the paper for a borderline acceptance.